# A Bayesian Model for Bivariate Causal Inference

**DOI:** 10.3390/e22010046

**Published:** 2019-12-29

**Authors:** Maximilian Kurthen, Torsten Enßlin

**Affiliations:** Max-Planck-Institut für Astrophysik, Karl-Schwarzschildstr. 1, 85748 Garching, Germany; ensslin@mpa-garching.mpg.de

**Keywords:** causal inference, bayesian model selection, information field theory, cause–effect pairs, additive noise

## Abstract

We address the problem of two-variable causal inference without intervention. This task is to infer an existing causal relation between two random variables, i.e., X→Y or Y→X, from purely observational data. As the option to modify a potential cause is not given in many situations, only structural properties of the data can be used to solve this ill-posed problem. We briefly review a number of state-of-the-art methods for this, including very recent ones. A novel inference method is introduced, *Bayesian Causal Inference* (*BCI*) which assumes a generative Bayesian hierarchical model to pursue the strategy of Bayesian model selection. In the adopted model, the distribution of the cause variable is given by a Poisson lognormal distribution, which allows to explicitly regard the discrete nature of datasets, correlations in the parameter spaces, as well as the variance of probability densities on logarithmic scales. We assume Fourier diagonal Field covariance operators. The model itself is restricted to use cases where a direct causal relation X→Y has to be decided against a relation Y→X, therefore we compare it other methods for this exact problem setting. The generative model assumed provides synthetic causal data for benchmarking our model in comparison to existing state-of-the-art models, namely *LiNGAM*, *ANM-HSIC*, *ANM-MML*, *IGCI*, and *CGNN*. We explore how well the above methods perform in case of high noise settings, strongly discretized data, and very sparse data. *BCI* performs generally reliably with synthetic data as well as with the real world *TCEP* benchmark set, with an accuracy comparable to state-of-the-art algorithms. We discuss directions for the future development of *BCI*.

## 1. Introduction

### 1.1. Motivation and Significance of the Topic

*Causal Inference* regards the problem of drawing conclusions about how some entity we can observe does—or does not—influence or is being influenced by another entity. Having knowledge about such law-like causal relations enables us to predict what will happen ($\hat{=}$ the effect) if we know how the circumstances ($\hat{=}$ the cause) do change. For example, one can draw the conclusion that a street will be wet (the effect) whenever it rains (the cause). Knowing that it will rain, or indeed observing the rainfall itself, enables one to predict that the street will be wet. Less trivial examples can be found in the fields of epidemiology (identifying some bacteria as the cause of a disease) or economics (knowing how taxes will influence the GDP of a country). Under ideal conditions the system under investigation can be manipulated. Such interventions might allow to set causal variables to specific values which allows to study their effects statistically. In many applications, like in astronomy, geology, global economics, and others, this is hardly possible. For example, in the area of astrophysics, observed properties of galaxies or galaxy clusters include redshift, size, spectral features, fluxes at different wavelengths for lines of sight through the Milky Way. There, an identification of causal directions has to rest on the hints the causal relation imprints onto the available data only. Restricting on the decision between a direct causal relation X→Y vs. Y→X ignores important possibilities (such as hidden confounders), however is still important as a decision within of a subset of possible causal relations and has been giving rise to own benchmark datasets [2].

Especially within the fields of data science and machine learning, specific tasks from causal inference have been attracting much interest recently. The authors of [3] propose that causal inference stands as a third main task of data science besides description and prediction. Reference [4] claims that the task of causal inference will be the next “big problem” for Machine Learning. Such a specific problem is the two variable causal inference, also addressed as the *cause–effect problem* by [5]. Given purely observational data from two random variables, *X* and *Y*, which are directly causally related, the challenge is to infer the correct causal direction. Interestingly, this is an incorporation of a fundamental asymmetry between cause and effect which does always hold and can be exploited to tackle such an inference problem. Given two random variables, *X* and *Y*, which are related causally, X→Y (“*X* causes *Y*”), there exists a fundamental independence between the distribution of the cause P(X) and the mechanism which relates the cause *X* to the effect *Y*. This independence, however, does not hold in the reverse direction. Most of the proposed methods for the inference of such a causal direction make use of this asymmetry in some way, either by considering the independence directly [2,6], or by taking into account the algorithmic complexity for the description of the factorization P(X)P(Y|X) and comparing it to the complexity of the reverse factorization P(Y)P(X|Y).

The point we want to make here is, that from the perspective of Bayesian probability theory, causal inference looks like a classical hypothesis testing problem. For this, the probability P(d|X→Y,M) is to be compared to P(d|Y→X,M). Here, d is the data, X→Y is the causal direction, and M is a (meta-) model of how causal relations between *X* and *Y* are typically realized. In the following we will adopt a specif choice for the model M. This choice and those of our numerical approximations could and should be criticized and eventually improved. The hypothesis we are following here, however, is that, given M, Bayesian inference is everything that is needed to judge causal directions. Different algorithms for causal inference might just result from different assumptions and different numerical approximations made.

### 1.2. Structure of the Work

The rest of the paper will be structured as follows. In Section 2 we will briefly outline and specify the problem setting. We also will review existing methods here, namely Additive Noise Models, Information Geometric Causal Inference, and Learning Methods. Section 3 will describe our inference model which is based on a hierarchical Bayesian model. In Section 4 we will accompany the theoretical framework with experimental results. To that end we outline the “forward model” which allows to sample causally related data in Section 4.1. We describe a specific algorithm for the inference model in Section 4.2, which is then tested on various benchmark data (Section 4.3). The performance is evaluated and compared to state-of-the-art methods mentioned in Section 2. We conclude in Section 5 by assessing that our model generally can show competitive classification accuracy and propose possibilities to further advance the model.

## 2. Problem Setting and Related Work

Here and in the following we assume two random variables, *X* and *Y*, which map onto measurable spaces X and Y. Our problem, the two-variable causal inference, is to determine if *X* causes *Y* or *Y* causes *X*, given only observations from these random variables. The possibilities that they are unrelated (X⊥⊥Y), connected by a confounder Z (X←Z→Y), or interrelated (X↔Y) are ignored here for the sake of clarity.

### 2.1. Problem Setting

Regarding the definition of causality we refer to the *do-calculus* introduced by [7]. Informally, the intervention do(X=x) can be described as setting the random variable *X* to attain the value *x*. This allows for a very compact definition of a causal relation X→Y (“*X* causes *Y*”) via
(1)$$\begin{array}{c} X\to Y\Leftrightarrow P\left(y|\mathrm{do}\left(x\right)\right)\neq P\left(y|\mathrm{do}\left(x^{\prime }\right)\right) \end{array}$$
for some $x,x^{\prime }$ being realizations of *X* and *y* being a realization of *Y* [2] (in general, the probabilities P(y|X=x) and P(y|do(X=x)) are different).

While the conditional probability P(y|X=x) corresponds to just observing the value *x* for the variable *X*, the do-probability P(y|do(X=x)) corresponds to a direct manipulation of the system, only modifying *x*, without changing any other variable directly.

We want to focus on the case of two observed variables, where either X→Y or Y→X holds. Our focus is on the specific problem to decide, in a case where two variables *X* and *Y* are observed, whether X→Y holds or Y→X We suppose to have access to a finite number of samples from the two variables, i.e., samples $x=(x_{1},\ldots ,x_{N})$ from *X* and $y=(y_{1},\ldots ,y_{N})$ from *Y*. Our task is to decide the true causal direction using only these samples:

**Problem** **1.**
*Prediction of causal direction for two variables*

***Input**: A finite number of sample data d≡(x,y), where $x=\left(x_{1},\ldots ,x_{N}\right),y=\left(y_{1},\ldots ,y_{N}\right)$*

***Output**: A predicted causal direction D∈{X→Y,Y→X}*


### 2.2. Related Work

Approaches to causal inference from purely observational data are often divided intothree groups [8,9], namely constraint-based, score-based, and asymmetry-based methods. Sometimes this categorization is extended by considering learning methods as a fourth, separate group. Constraint-based and score-based methods are using conditioning on external variables. In a two-variable case there are no external variables, so they are of little interest here.

Asymmetry-based methods exploit an inherent asymmetry between cause and effect. This asymmetry can be framed in different terms. We will follow a similar overview here as [5], where we also refer to for a more detailed discussion. One way is to use the concept of algorithmic complexity—given a true direction X→Y, the factorization of the joint probability into P(X,Y)=P(X)P(Y|X) will be less complex than the reverse factorization P(Y)P(X|Y). Such an approach is often used by *Additive Noise Models* (ANMs). This family of inference models assume additive noise, i.e., in the case X→Y, *Y* is determined by some function *f*, mapping *X* to *Y*, and some collective noise variable $E_{Y}$, i.e., $Y=f\left(X\right)+E_{Y}$, where *X* is independent of $E_{Y}$.

An early model called *LiNGAM* [10] uses Independent Component Analysis on the data belonging to the variables. This model however makes the assumptions of linear relations and non-Gaussian noise.

A more common approach is to use some kind of regression (e.g., Gaussian process regression) to get an estimate on the function *f* and measure how well the model such obtained fits the data. The latter is done by measuring independence between the cause variable and the regression residuum (*ANM-HSIC*, [2,11]), or by employing a Bayesian model selection (introduced besides other methods as (*ANM-HSIC*, [12]).

Another way of framing the asymmetry mentioned above is to state that the mechanism relating cause and effect should be independent of the cause. This formulation is employed by the concept of *IGCI* (*Information Geometric Causal Inference*, [6]).

The recent advances in the field of deep learning are represented in an approach called *CGNN* (*Causal Generative Neural Networks*, [13]). The authors use Generative Neural Networks to model the distribution of one variable given samples from the other variable. As Neural Networks are able to approximate nearly arbitrary functions, the direction where such a artificial modelling is closer to the real distributions (inferred from the samples) is preferred.

Finally, *KCDC* (*Kernel Conditional Deviance for Causal Inference*, [9]) uses the thought of asymmetry in the algorithmic complexity directly on the conditional distributions P(X|Y=y),P(Y|X=x). The model measures the variance of the conditional mean embeddings of the above distributions and prefers the direction with the less varying embedding.

A somewhat related approach employed by [14] uses a Bayesian model in combination with MCMC sampling in order to reconstruct Bayesian networks. The authors of [15] use a Bayesian approach specifically in the domain of causal model discovery and compare it to constraint-based approaches. However, an approach which considers multi-variable causal models is fundamentally different from a 2-variable scenario (as conditioning on a third variable is not possible in the latter case).

## 3. A Bayesian Inference Model

Our contribution incorporates the concept of Bayesian model selection and builds on the formalism of *Information Field Theory* (*IFT, the information theory for fields* [16]). Parts of this paper are taken from the master’s thesis of one of the authors [1]. Bayesian model selection compares two competing models, in our case X→Y and Y→X, and asks for the ratio of the marginalized likelihoods,
$$\begin{array}{c} O_{X\to Y}=\frac{P(d|X\to Y,M)}{P(d|Y\to X,M)}, \end{array}$$
the so called *Bayes Factor*. Here, *M* denotes the hyperparameters which are assumed to be the same for both models and are yet to be specified.

In the setting of the present causal inference problem, a similar approach has already been used by [12]. This approach however does use a Gaussian mixture model for the distribution of the cause variable while we model the logarithmic cause distribution as a more flexible Gaussian random field (or Gaussian process) [17] and explicitly consider an additional discretization via introducing counts in bins (using a Poissonian statistic on top). Gaussian random fields have, in principle, an infinite number of degrees of freedom, making them an interesting choice to model our distribution of the cause variable and the function relating cause and effect. The formalism of *IFT* borrows computational methods from quantum field theory for computations with such random fields.

Throughout the following we will consider X→Y as the true underlying direction which we derive our formalism for. The derivation for Y→X will follow analogously by switching the variables. We will begin with deriving in Section 3.1 the distribution of the cause variable, P(X|X→Y,M). In Section 3.2 we continue by considering the conditional distribution P(Y|X,X→Y,M). Combining those results, we compute then the full Bayes factor in Section 3.3.

### 3.1. Distribution of the Cause Variable

Without imposing any constraints, we reduce our problem to the interval [0,1] by assuming that X=Y=[0,1]. This can always be ensured by rescaling the data. We make the assumption that in principle the cause variable *X* follows a lognormal distribution.
$$\begin{array}{c} P\left(x|\beta \right)\propto e^{\beta (x)} \end{array}$$
with $\beta \in R^{\left[0,1\right]}$ (throughout the paper we will use the set theory notation for functions, i.e., for a function f:X→Y we write $f\in Y^{X}$, which allows a concise statement of domain and codomain without imposing any further restrictions), being some signal field which follows a zero-centered normal distribution, β∼N(β|0,B).

Here we write *B* for the covariance operator $E_{\beta ∼P(\beta )}\left[\beta \left(x_{0}\right)\beta \left(x_{1}\right)\right]=B\left(x_{0},x_{1}\right)$.

We postulate statistical homogeneity for the covariance, that is
$$\begin{array}{c} E_{\beta ∼P(\beta )}\left[\beta \left(x\right)\right]=E\left[\beta \left(x+t\right)\right]\\ E_{\beta ∼P(\beta )}\left[\beta \left(x\right)\beta \left(y\right)\right]=E\left[\beta \left(x+t\right)\beta \left(y+t\right)\right] \end{array}$$
i.e., first and second moments should be independent on the absolute location. The *Wiener–Khintchine Theorem* now states that the covariance has a spectral decomposition, i.e., it is diagonal in Fourier space, under this condition (see, e.g., [18]). Denoting the Fourier transform by F, i.e., in the one dimensional case, $F\left[f\right]\left(q\right)=\int dx\;e^{-iqx}f\left(x\right)$. Therefore, the covariance can be completely specified by a one dimensional function:$$\begin{array}{c} \left(FBF^{-1}\right)\left(k,q\right)=2\pi \delta \left(k-q\right)P_{\beta }\left(k\right) \end{array}$$

Here, $P_{\beta }\left(k\right)$ is called the *power spectrum*.

Building on these considerations we now regard the problem of discretization. Measurement data itself is usually not purely continuous but can only be given in a somewhat discretized way (e.g., by the measurement device itself or by precision restrictions imposed from storing the data). Another problem is that many numerical approaches to inference tasks, such as Gaussian Process regression, use finite bases as approximations in order to efficiently obtain results [2,12]. Here, we aim to directly confront these problems by imposing a formalism where the discretization is inherent.

So instead of taking a direct approach with the above formulation, we use a Poissonian approach and consider an equidistant grid $\left\{z_{1},\ldots ,z_{n_{\mathrm{bins}}}\right\}$ in the [0,1] interval. This is equivalent to defining bins, where the $z_{j}$ are the midpoints of the bins. We now take the measurement counts, $k_{i}$, which gives the number of *x*-measurements within the *i*-th bin. For these measurement counts we now take a Poisson lognormal distribution as an Ansatz, that is, we assume that the measurement counts for the bins are Poisson distributed, where the means follow a lognormal distribution. We argue that this is indeed a justified approach here, as in a discretized scenario we have to deal with count-like integer data (where a Poisson distribution is natural). The lognormal distribution of the Poisson parameters is in our eyes well-justified here. On the one hand, it takes into account the non-negativity of the Poisson parameter. On the other hand, only proposing a normal distribution of the log allows for a uncertainty in the order of magnitude while permitting for spatial correlation in this log density.

We can model this discretization by applying a response operator $R:R^{\left[0,1\right]}\to R^{n_{\mathrm{bins}}}$ to the lognormal field. This is done in the most direct way via employing a Dirac delta distribution
$$\begin{array}{cc} \; & R_{jx}\equiv \delta \left(x-z_{j}\right) \end{array}$$

In order to allow for a more compact notation we will use an index notation from now on, e.g., $f_{x}=f\left(x\right)$ for some function *f* or $O_{xy}=O\left(x,y\right)$ for some operator *O*. Whenever the indices are suppressed, an integration (in the continuous case) or dot product (in the discrete case) is understood, e.g., ${\left(Of\right)}_{x}\equiv O_{xy}f_{y}=\int dyO_{xy}f_{y}=\int dyO\left(x,y\right)f\left(y\right)$ In the following we will use bold characters for finite dimensional vectors, e.g., $\lambda \equiv {\left(\lambda_{1},\ldots ,\lambda_{n_{\mathrm{bins}}}\right)}^{T}$. By inserting such a finite dimensional vector in the argument of a function, e.g., β(x) we refer to a vector consisting of the function evaluated at each entry of x, that is $\beta \left(z\right)\equiv (\beta \left(z_{1}\right),\ldots ,\beta \left(z_{n_{\mathrm{bins}}}\right)$. Later on we will use the notation $\hat{\cdot }$ which raises some vector to a diagonal matrix (${\hat{x}}_{ij}\equiv \delta_{ij}x_{i}$ (no summation implicated)). We will use this notation analogously for fields, e.g., (${\hat{\beta }}_{uv}\equiv \delta \left(u-v\right)\beta \left(u\right)$). Writing $1^{†}R$ denotes the dot product of the vector R with a vector of ones and hence corresponds to the summation of the entries of R ($1^{†}R=\sum_{j}R_{j}$). Now we can state the probability distribution for $k_{j}$, the measurement count in bin *j*:$$\begin{array}{cc} \; & P\left(k_{j}|\lambda_{j}\right)=\frac{\lambda_{j}^{k_{j}}e^{-\lambda_{j}}}{k_{j}!}\\ \; & \lambda_{j}=E_{\left(k|\beta \right)}\left[k_{j}\right]=\rho e^{\beta_{z_{j}}}=\int dxR_{jx}e^{\beta_{x}}=\rho {\left(Re^{\beta }\right)}_{j} \end{array}$$
Therefore, considering the whole vector λ of bin means and the vector k of bin counts at once:$$\begin{array}{cc} \; & \lambda =\rho Re^{\beta }=\rho e^{\beta (z)}\\ \; & P\left(k|\lambda \right)=\prod_{j}\frac{\lambda_{j}^{k_{j}}e^{-\lambda_{j}}}{k_{j}!}=\prod_{j}\frac{{\left(R_{j}e^{\beta }\right)}^{k_{j}}e^{-R_{j}e^{\beta }}}{k_{j}!}=\frac{\left(\prod_{j}{\left(R_{j}e^{\beta }\right)}^{k_{j}}\right)e^{-1^{†}Re^{\beta }}}{\prod_{j}k_{j}!}\\ \; & P\left(x|k\right)=\frac{1}{N!} \end{array}$$

The last equation follows from the consideration that given the counts $\left(k_{1},\ldots ,k_{n_{\mathrm{bins}}}\right)$ for the bins, only the positions of the observations $\left(x_{1},\ldots ,x_{N}\right)$ is fixed, but the ordering is not. The *N* observations can be ordered in N! ways.

Now considering the whole vector of bin counts k at once, we get
(2)$$\begin{array}{cc} \; & P\left(k|\beta \right)=\frac{e^{\sum_{j}k_{j}\beta \left(z_{j}\right)}e^{-\rho^{†}e^{\beta (z)}}}{\prod_{j}k_{j}!}=\frac{e^{k^{†}\beta \left(z\right)-\rho^{†}e^{\beta (z)}}}{\prod_{j}k_{j}!} \end{array}$$

A marginalization in β involving a Laplace approximation around the most probable $\beta =\beta_{0}$ leads to (see Appendix A for a detailed derivation):(3)$$\begin{array}{cc} \; & P\left(x|P_{\beta },X\to Y\right)\approx \frac{1}{N!}\frac{e^{+k^{†}\beta_{0}-\rho^{†}e^{\beta_{0}}-\frac{1}{2}\beta_{0}^{†}B^{-1}\beta_{0}}}{{\left|\rho B\hat{e^{\beta_{0}}}+𝟙\right|}^{\frac{1}{2}}\prod_{j}k_{j}!} \end{array}$$(4)$$\begin{array}{cc} \; & H\left(x|P_{\beta },X\to Y\right)\approx H_{0}+\frac{1}{2}\log\left|\rho B\hat{e^{\beta_{0}}}+\hat{𝟙}\right|+\log\left(\prod_{j}k_{j}!\right)-k^{†}\beta_{0}+\rho^{†}e^{\beta_{0}}+\frac{1}{2}\beta_{0}^{†}B^{-1}\beta_{0} \end{array}$$
where H(·)≡−log(P(·)) is called the *information Hamiltonian* in *IFT* and $H_{0}$ collects all terms which do not depend on the data d.

### 3.2. Functional Relation of Cause and Effect

Similarly to β, we suppose a Gaussian distribution for the function *f*, relating *Y* to *X*:$$\begin{array}{c} R^{\left[0,1\right]}∋f∼N\left(0|f,F\right) \end{array}$$

Proposing a Fourier diagonal covariance *F* once more, determined by a power spectrum $P_{f}$,
$$\begin{array}{c} \left(FFF^{-1}\right)\left(k,q\right)=2\pi \delta \left(k-q\right)P_{f}\left(k\right), \end{array}$$
we assume additive Gaussian noise, using the notation $f\left(x\right)\equiv {\left(f\left(x_{1}\right),\ldots ,f\left(x_{N}\right)\right)}^{T}$ and $\epsilon \equiv {\left(\epsilon_{1},\ldots ,\epsilon_{N}\right)}^{T}$.

So, in essence we are proposing a Gaussian Process Regression with a Fourier diagonal covariance. We have
(5)$$\begin{array}{cc} y & =f(x)+\epsilon \\ \epsilon & ∼N(\epsilon |0,E)\\ E & \equiv \mathrm{diag}\left(\varsigma^{2},\varsigma^{2},\ldots \right)=\varsigma^{2}𝟙\in R^{N\times N}, \end{array}$$
that is, each independent noise sample is drawn from a zero-mean Gaussian distribution with given variance $\varsigma^{2}$.

Knowing the noise ϵ, the cause x and the causal mechanism *f* completely determines y via Equation (5). Therefore, P(y|x,f,ϵ,X→Y)=δ(y−f(x)−ϵ). We can now state the conditional distribution for the effect variable measurements y, given the cause variable measurements x. Marginalizing out the dependence on the relating function *f* and the noise ϵ we get:(6)$$\begin{array}{cc} P(y|x,P_{f},\varsigma ,X\to Y) & =\int \frac{d^{N}q}{{\left(2\pi \right)}^{N}}e^{iq^{†}y-\frac{1}{2}q\left(\tilde{F}+E\right)q}\\ \; & ={\left(2\pi \right)}^{-\frac{N}{2}}{\left|\tilde{F}+E\right|}^{-\frac{1}{2}}e^{-\frac{1}{2}y^{†}{\left(\tilde{F}+E\right)}^{-1}y} \end{array}$$

In the equation above, $\tilde{F}$ denotes a the N×N-matrix with entries ${\tilde{F}}_{ij}=F\left(x_{i},x_{j}\right)$ (this type of matrix, i.e., the evaluation of covariance or kernel at certain positions, is sometimes called a Gram matrix). Again, we give a detailed computation in the Appendix A.

### 3.3. Computing the Bayes Factor

Now we are able to calculate the full likelihood of the data d=(x,y) given our assumptions $P_{\beta },P_{f},\varsigma$ for the direction X→Y and vice versa Y→X (via the full model as given in Figure 1 As we are only interested in the ratio of the probabilities and not in the absolute probabilities itself, it suffices to calculate the Bayes factor:$$\begin{array}{cc} O_{X\to Y} & =\frac{P(d|P_{\beta },P_{f},\varsigma ,X\to Y)}{P(d|P_{\beta },P_{f},\varsigma ,Y\to X)}\\ \; & =\exp[H\left(d|P_{\beta },P_{f},\varsigma ,Y\to X\right)-H\left(d|P_{\beta },P_{f},\varsigma ,X\to Y\right)] \end{array}$$

Here we used again the information Hamiltonian H(·)≡−logP(·)

Making use of Equations (3) and (6) we get, using the calculus for conditional distributions on the Hamiltonians, H(A,B)=H(A|B)+H(B),
(7)$$\begin{array}{cc} H(d|P_{\beta },P_{f},\varsigma ,X\to Y) & =H\left(x|P_{\beta },X\to Y\right)+H\left(y|x,P_{f},\varsigma ,X\to Y\right)\\ \; & =H_{0}+\log\left(\prod_{j}k_{j}!\right)+\frac{1}{2}\log\left|\rho B\hat{e^{\beta_{0}}}+𝟙\right|-k^{†}\beta_{0}+\\ \; & +\rho^{†}e^{\beta_{0}}+\frac{1}{2}\beta_{0}^{†}B^{-1}\beta_{0}+\frac{1}{2}y^{†}{\left(\tilde{F}+E\right)}^{-1}y+\frac{1}{2}\left|\tilde{F}+E\right| \end{array}$$

In this formula we suppressed the dependence of $\tilde{F},\beta_{0}$ on x (for the latter, the dependence is not explicit, but rather implicit as $\beta_{0}$ is determined by the minimum of the x-dependent functional γ). We omit stating $H(d|P_{\beta },P_{f},\varsigma ,Y\to X)$ explicitly as the expression is just given by taking Equation (7) and switching x and y or *X* and *Y*, respectively.

## 4. Implementation and Benchmarks

We can use our model in a forward direction to generate synthetic data with a certain underlying causal direction. We describe this process in Section 4.1. In Section 4.2 we give an outline on the numerical implementation of the inference algorithm. This algorithm is tested on and compared on benchmark data. To that end we use synthetic data and real world data. We describe the specific datasets and give the results in Section 4.3.

### 4.1. Sampling Causal Data via a Forward Model

To estimate the performance of our algorithm and compare it with other existing approaches, a benchmark dataset is of interest to us. Such benchmark data is usually either real world data or synthetically produced. While we will use the *TCEP* benchmark set of [2] in Section 4.3.2, we also want to use our outlined formalism to generate artificial data representing causal structures. Based on our derivation for cause and effect we implement a forward model (1) to generate data d as following Algorithm 1.

**Algorithm 1** Sampling of causal data via forward model
**Input:** Power spectra $P_{\beta },P_{f}$, noise variance $\varsigma^{2}$, number of bins $n_{\mathrm{bins}}$, desired number (As we draw the number of samples from Poisson distribution in each bin, we do not deterministically control the total number of samples) of samples $\tilde{N}$**Output:***N* samples $\left(d_{i}\right)=\left(x_{i},y_{i}\right)$ generated from a causal relation of either X→Y or Y→XDraw a sample field $\beta \in R^{\left[0,1\right]}$ from the distribution N(β|0,B)Set an equally spaced grid with $n_{\mathrm{bins}}$ points in the interval [0,1]: $z=\left(z_{1},\ldots ,z_{n_{\mathrm{bins}}}\right),z_{i}=\frac{i-0.5}{n_{\mathrm{bins}}}$Calculate the vector of Poisson means $\lambda =(\lambda_{1},\ldots \lambda_{n_{\mathrm{bins}}})$ with $\lambda_{i}\propto e^{\beta (z_{i})}$At each grid point $i\in \{1,\ldots ,n_{\mathrm{bins}}\}$, draw a sample $k_{i}$ from a Poisson distribution with mean $\lambda_{i}$: $k_{i}∼P_{\lambda_{i}}\left(k_{i}\right)$Set $N=\sum_{i=1}^{n_{\mathrm{bins}}}k_{i}$For each $i\in \{1,\ldots ,n_{\mathrm{bins}}\}$ add $k_{i}$ times the element $z_{i}$ to the set of measured $x_{j}$. Construct the vector $x=(\ldots ,\underset{k_{i}\;\mathrm{times}}{\underset{︸}{z_{i},z_{i},z_{i}}},\ldots )$Draw a sample field $f\in R^{\left[0,1\right]}$ from the distribution N(f|0,F). Rescale *f* s.th. $f\in {\left[0,1\right]}^{\left[0,1\right]}$Draw a multivariate noise sample $\epsilon \in R^{N}$ from a normal distribution with zero mean and variance $\varsigma^{2}$, $\epsilon ∼N(\epsilon |0,\varsigma^{2})$Generate the effect data y by applying *f* to x and adding ϵ: y=f(x)+ϵWith probability $\frac{1}{2}$ return $d=(x^{T},y^{T})$, otherwise return $d=(y^{T},x^{T})$


Comparing the samples for different power spectra (see Figure 2), we decide to sample data with power spectra $P\left(q\right)=\frac{1000}{q^{4}+1}$ and $P\left(q\right)=\frac{1000}{q^{6}+1}$, as these seem to resemble “natural” mechanisms, see Figure 2.

### 4.2. Implementation of the Bayesian Causal Inference Model

Based on our derivation in Section 3 we propose a specific algorithm to decide the causal direction of a given dataset and therefore give detailed answer for Problem 1. Basically, the task comes down to find the minimum $\beta_{0}$ for the saddle point approximation and calculate the terms given in Equation (7):

We provide an implementation of Algorithm 2 in *Python* (https://gitlab.mpcdf.mpg.de/ift/bayesian_causal_inference). We approximate the operators B,F as matrices $\in R^{n_{\mathrm{bins}}\times n_{\mathrm{bins}}}$, which allows us to explicitly numerically compute the determinants and the inverse. As the most critical part we consider the minimization of β, i.e., step 4 in Algorithm 2. As we are however able to analytically give the curvature $\Gamma_{\beta }$ and the gradient $\partial_{\beta }\gamma$ of the energy γ to minimize, we can use a Newton-scheme here. We derive satisfying results (see Figure 3) using the *Newton-CG* algorithm [19], provided by the *SciPy*-Library [20]. After testing our algorithm on different benchmark data, we choose the default hyperparameters as
(8)$$\begin{array}{cc} \; & P_{\beta }=P_{f}\propto \frac{1}{q^{4}+1}, \end{array}$$
(9)$$\begin{array}{cc} \; & \varsigma^{2}=0.01, \end{array}$$
(10)$$\begin{array}{cc} \; & r=512, \end{array}$$
(11)$$\begin{array}{cc} \; & \rho =1. \end{array}$$

**Algorithm 2** 2-variable causal inference
**Input:** Finite sample data $d\equiv \left(x,y\right)\in R^{N\times 2}$, Hyperparameters $P_{\beta },P_{f},\varsigma^{2},r$**Output**: Predicted causal direction $D_{X\to Y}\in \left\{X\to Y,Y\to X\right\}$Rescale the data to the [0,1] interval. That is, $\min\left\{x_{1},\ldots ,x_{N}\right\}=\min\left\{y_{1},\ldots ,y_{N}\right\}=0$ and $\max\left\{x_{1},\ldots ,x_{N}\right\}=\max\left\{y_{1},\ldots ,y_{N}\right\}=1$Define an equally spaced grid of $\left(z_{1},\ldots ,z_{n_{\mathrm{bins}}}\right)$ in the interval [0,1]Calculate matrices B,F representing the covariance operators *B* and *F* evaluated at the positions of the grid, i.e., $B_{ij}=B\left(z_{i},z_{j}\right)$Find the $\beta_{0}\in R^{\left[0,1\right]}$ for which γ, as defined in Section A.1 (Equation (A2)), becomes minimalCalculate the d-dependent terms of the information Hamiltonian in Equation (7) (i.e., all terms except $H_{0}$)Repeat steps 4 and 5 with y and x switchedCalculate the Bayes factor $O_{X\to Y}$If $O_{X\to Y}>1$, return X→Y, else return Y→X


While fixing the power spectra might seem somewhat arbitrary, we remark that this corresponds to fixing a kernel, e.g., as a squared exponential kernel, which is done in many publications (e.g., [9,13]).

Future extensions of our method might learn $P_{\beta }$ and $P_{f}$ if the data is rich enough.

### 4.3. Benchmark Results

We compare our outlined model, in the following called *BCI* (*Bayesian Causal Inference*), to a number of state-of-the-art approaches. The selection of the considered methods is influenced by the ones in recent publications, e.g., [9,13]. Namely, we include the *LiNGAM* algorithm, acknowledging it as one of the oldest models in this field and a standard reference in many publications. We also use the *ANM* Algorithm [2] with HSIC and Gaussian Process Regression (*ANM-HSIC*) as well as the *ANM-MML* approach [12]. The latter uses a Bayesian Model Selection, arguably the closest to the algorithm proposed in this publication, at least to our best knowledge. We further include the *IGCI* algorithm, as it differs fundamentally in its formulation from the ANM algorithms and has shown strong results in recent publications [2,9,13]. We employ the IGCI algorithm with entropy estimation for scoring and a Gaussian distribution as reference distribution.

Finally, *CGNN* [13] represents the rather novel influence of deep learning methods. We use the implementation provided by the authors, with itself uses *Python* with the *Tensorflow* [21] library. The most critical hyper-parameter here is, as the authors themselves mention, the number of hidden neurons which we set to a value of $n_{h}=30$, as this is the default in the given implementation and delivers generally good results. We use 32 runs each, as recommended by the authors of the algorithm.

A comparison with the *KCDC* algorithm would be interesting, unfortunately the authors did not provide any computational implementation so far (October 2019). We compare the mentioned algorithms to *BCI* on basis of synthetic and real world data. For the synthetic data we use our forward model, as outlined in Section 4.1 with varying parameters. For the real world data we use the well-known *TCEP* dataset (*Tuebingen Cause Effect Pairs,* [2]).

#### 4.3.1. Results for Synthetic Benchmark Data

We generate our synthetic data adopting the power spectra $P\left(q\right)=\frac{1}{q^{4}+1}$ for both, $P_{\beta }$, $P_{f}$. We further set $n_{\mathrm{bins}}$ = 512, $\tilde{N}$ = 300, and $\varsigma^{2}$ = 0.05 as default settings. We provide the results of the benchmarks in Table 1. While *BCI* achieves almost perfect results (98%), the assessed *ANM* algorithms provide perfect performance here. As a first variation, we explore the influence of high and very high noise on the performance of the inference models. Therefore, we set the parameter $\varsigma^{2}$ = 0.2 for high noise and $\varsigma^{2}$ = 1 for very high noise in Algorithm 1, while keeping the other parameters set to the default values. While our *BCI* algorithm is affected but still performs reliably with an accuracy of ≥90%, the *ANM* algorithms are remarkably robust in the presence of the noise. This is likely due to the fact that the distribution of the true cause P(X) is not influenced by high noise and this distribution is assessed on its own by those.

As our model uses a Poissonian approach, which explicitly considers discretization effects of data measurement, it is of interest how the performance behaves when using a strong discretization. We emulate such a situation by employing our forward model with a very low number of bins. Again, we keep all parameters to default values and set $n_{\mathrm{bins}}$ = 16 and $n_{\mathrm{bins}}$ = 8 for synthetic data with high and very high discretization. The *ANM* models again turn out to be robust again discretization. *CGNN* and *IGCI* perform significantly worse here. In the case of *IGCI* this can be explained by the entropy estimation, which simply removes non-unique samples. Our *BCI* algorithm is able to achieve over 90% accuracy here.

We explore another challenge for inference algorithms by strongly reducing the number of samples. While we sampled about 300 observations with our other forward models so far, here we reduce the number of observed samples to 30 and 10 samples. In this case *BCI* performs very well compared to the other models, in fact it is able to outperform them in the case of just 10 samples being given. We note that of course *BCI* does have the advantage that it “knows” the hyperparameters of the underlying forward model. Yet we consider the results as encouraging, and this advantage will be removed in the confrontation with real world data.

#### 4.3.2. Results for Real World Benchmark Data

The most widely used benchmark set with real world data is the *TCEP* [2]. We use the 102 2-variable datasets from the collection with weights as proposed by the maintainers. As some of the contained datasets include a high number of samples (up to 11,000), we randomly subsample large datasets to 500 samples each in order to keep computation time maintainable. We did not include the *LiNGAM* algorithm here, as we experienced computational problems with obtaining results here for certain datasets (namely pair0098). The authors of [13] report the accuracy of *LiNGAM* on the *TCEP* dataset to be around 40%. *BCI* performs generally comparable to established approaches as *ANM* and *IGCI*. *CGNN* performs best with an accuracy about 70% here, a bit lower than the one reported by [13] of around 80%. The reason for this is arguably to be found in the fact that we set all hyperparameters to fixed values, while [13] used a leave-one-out-approach to find the best setting for the hyperparameter $n_{h}$.

Motivated by the generally strong performance of our approach in the case of sparse synthetic data, we also explore a situation where real world data is only sparsely available. To that end, we subsample all *TCEP* datasets down to 75 randomly chosen samples kept for each one. To circumvent the chance of an influence of the subsampling procedure we average the results over 20 different subsamplings. The results are as well given in Table 2. The loss in accuracy of our model is rather small.

## 5. Discussion

The problem of purely bivariate causal discovery is a rather restricted one as it ignores the possibility of causal independence or hidden confounders. However, it is a fundamental one and still remains as a part of the decision within subsets of possible causal structures. It also might be a valuable contribution to real world problems such as in astrophysics, where one might be interested in discovering the main causal direction within the multitude of discovered variables. The Bayesian Causal Inference method introduced builds on the formalism of information field theory. In this regard, we employed the concept of Bayesian model selection and made the assumption of additive noise, i.e., x=f(y)+ϵ. In contrast to other methods which do so, such as *ANM-MML*, we do not model the cause distribution by a Gaussian mixture model but by a Poisson Lognormal statistic.

We could show that our model is able to provide classification accuracy in the present causal inference task that is comparable to the one of other methods in the field. Our method is of course restricted to a bivariate problem, i.e., determining the direction of a causal relation between two one-dimensional variables. One difference from our model to existing ones is arguably to be found in the choice of the covariance operators. While our method uses, at its heart, Gaussian Process Regression, most other publications which do so use squared exponential kernels. We, however, choose a covariance which is governed by a $\frac{1}{q^{4}+1}$ power spectrum. This permits detecting more structures at small scales than methods using a squared exponential kernel.

As a certain weak point of *BCI*, we consider the approximation of the uncomputable path integrals via the Laplace approximation. A thorough investigation of error bounds (e.g., [22]) is yet to be carried out. As an alternative, one can think about sampling-based approaches to approximate the integrals. A recent publication [23] introduced a harmonic mean-based sampling approach to approximate moderate dimensional integrals. Adopting such a technique to our very high dimensional case might be promising to improve *BCI*. Furthermore, the novel *Metric Gaussian Variational Inference method* (MGVI [24]) might allow to go beyond the saddle point approximation method used here. At this point, we also want to mention that our numerical implementation will have difficulty to scale well to very large (lots of data points) problems. This is, however, also an issue for competing models and a subsampling procedure can always be used to decrease the scale of the problem.

Another interesting perspective is provided by deeper hierarchical models. While the outlined method took the power spectra and the noise variance as fixed hyperparameters, it would also be possible to infer these as well in an extension of the method. *MGVI* has already permitted the inference of rather complex hierarchical Bayeisan models [25]. Yet, the implementation of our model with fixed noise variance and power spectra was able to deliver competitive results with regard to state-of-the-art methods in the benchmarks. In particular, our method seems to be slightly superior in the low sample regime, probably due to the more appropriate Poisson statistic used. We consider this as an encouraging result for a first work in the context of information field theory-based causal inference.

## Figures and Tables

**Figure 1 entropy-22-00046-f001:**
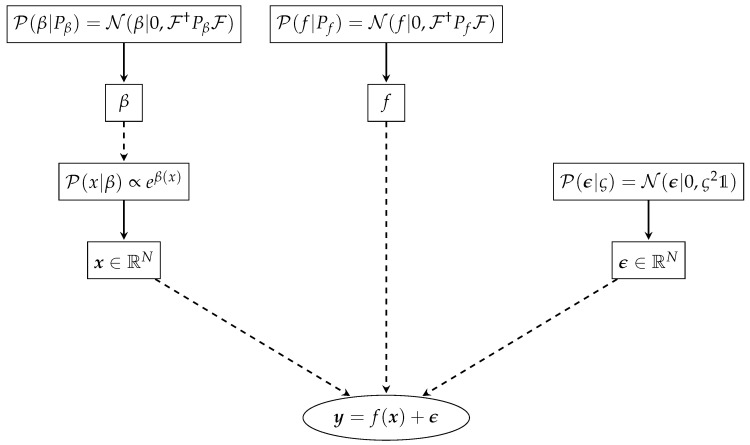
Overview over the used Bayesian hierarchical model, for the case X→Y.

**Figure 2 entropy-22-00046-f002:**
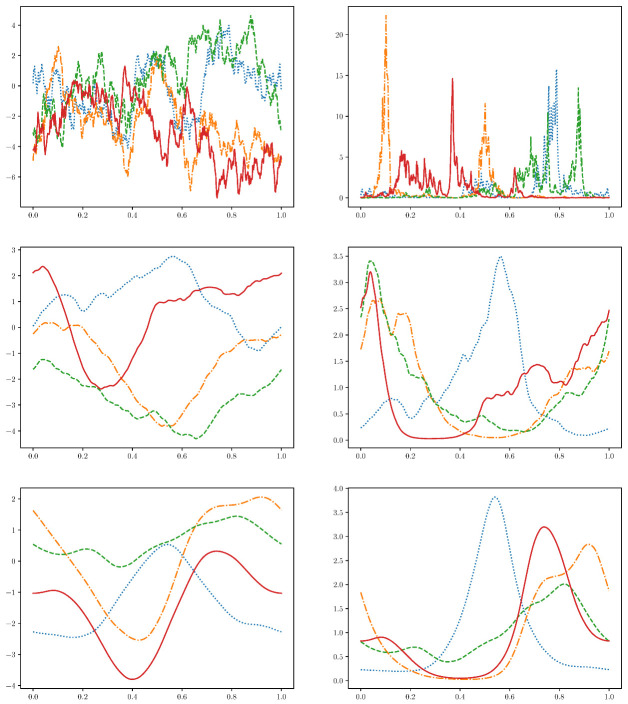
Different field samples from the distribution $N(\cdot |0,F^{†}\hat{P}F)$ (on the left) with the power spectrum $P\left(q\right)\propto \frac{1}{q^{2}+1}$ (top), $P\left(q\right)\propto \frac{1}{q^{4}+1}$ (middle), $P\left(q\right)\propto \frac{1}{q^{6}+1}$ (bottom). On the left, the field values themselves are plotted, on the right an exponential function is applied to those and the fields are normalized, i.e., as in our formulation $\lambda_{j}\propto \frac{e^{\beta (z_{j})}}{\int dze^{\beta (z)}}$ (Same colors/line styles on the right and the left indicate the same underlying functions (colors itself chosen just for distinguishability).

**Figure 3 entropy-22-00046-f003:**
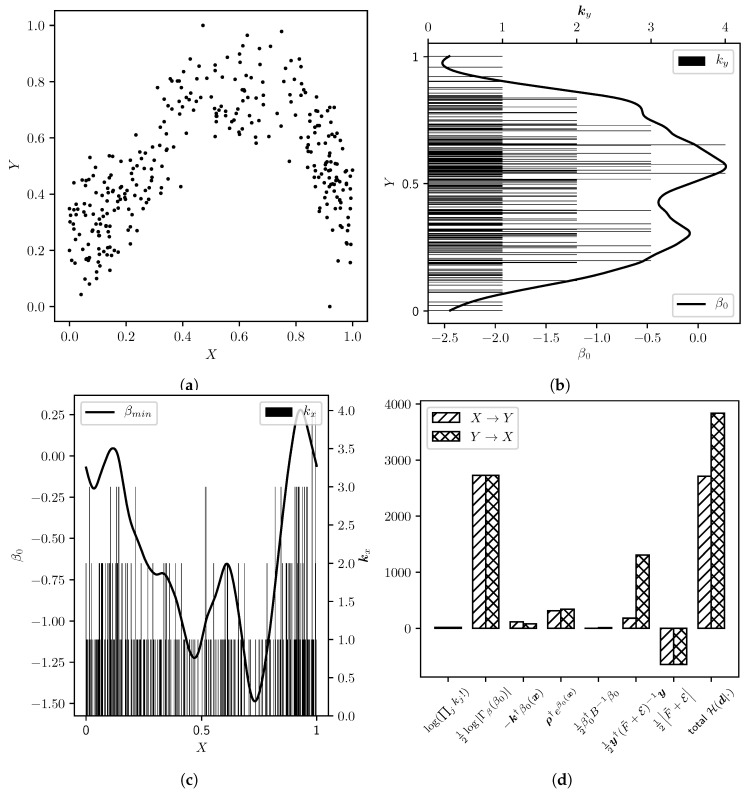
Illustration of a *Bayesian Causal Inference* run on synthetic data generated for causality X→Y. Here, the method clearly favours this causality with an odds ratio of $O_{X\to Y}\approx 10^{500}:1$. (**a**) Synthetic data, with causality X→Y; (**b**) Count histogram (k) and inferred $\beta_{0}$ for the model in the direction Y→X; (**c**) Count histogram (k) and inferred $\beta_{0}$ for the model in the direction X→Y; (**d**) Values of terms in $H(d|P_{\beta },P_{f},\varsigma ,X\to Y)$ and $H(d|P_{\beta },P_{f},\varsigma ,Y\to X)$. Smaller values increase the probability of the respective direction.

**Table 1 entropy-22-00046-t001:** Accuracy for the synthetic data benchmark. All parameters for the forward model besides the mentioned one are kept to default values, namely $n_{\mathrm{bins}}=512$, $\tilde{N}=300$, $\varsigma^{2}=0.05$.

Model	Default	$\varsigma^{2}$ = 0.2	$\varsigma^{2}$ = 1	$n_{\mathrm{bins}}$ = 16	$n_{\mathrm{bins}}$ = 8	30 Samples	10 Samples
BCI	0.98	0.94	0.90	0.93	0.97	0.92	0.75
LiNGAM	0.30	0.31	0.40	0.23	0.21	0.44	0.45
ANM-HSIC	1.00	0.98	0.94	0.99	1.00	0.91	0.71
ANM-MML	1.00	0.99	0.99	1.00	1.00	0.98	0.69
IGCI	0.65	0.60	0.58	0.24	0.09	0.48	0.40
CGNN	0.72	0.75	0.77	0.57	0.22	0.46	0.39

**Table 2 entropy-22-00046-t002:** Accuracy for TCEP Benchmark.

Model	TCEP	TCEP with 75 Samples
BCI	0.64	0.60
ANM-HSIC	0.63	0.54
ANM-MML	0.58	0.56
IGCI	0.66	0.62
CGNN	0.70	0.69

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
