# Peer review of "A Bayesian Model for Bivariate Causal Inference"

_entropy, 2019, doi:10.3390/e22010046_

Round 1

Reviewer 1 Report

Dear Authors,

your manuscript 'Bayesian Causal Inference' addresses a relevant topic - how to assess a causal relation from observations only. The paper focusses on the bivariate case.

The title of the manuscript is misleading: It raises the impression that the manuscript constitutes a review article on that topic. However, the contribution is a new algorithm to  the problem of causal inference in the special case of two random variables. The title should reflect this. Furthermore, precisely the topic of bivariate causal inference has been addressed extensively in the book 'Elements of Causal Inference' (2017) by J. Peters et al. However, this book is mentioned only very briefly in the introduction and not referred to at all in the section on 'problem setting and related work'. This is very unfortunate because the reader would benefit from the more elaborate discussion of the underlying concepts there and could better relate to the manuscript.

Which brings me to the main concerns:

a) Some parts of the paper need to be improved significantly on the notation style being used. I got really lost in section 3.1. Which quantities are vectors, which ones are fields, what denotes the range of a parameter, and what are domain definitions, etc. . It may be obvious to the authors but definitely not to the readership. Just an example from section 3.1, line 138: with beta element of R^[0,1] ..some signal field which follows beta~Normal(beta|0,B): First thought: beta is a scalar which takes values between zero and one. Then: beta is a real function on the line [0,1]; next: beta is a Gaussian process. Looking back: wait, there is exp(beta(x))... Being more explicit and precise would spare the reader the decoding process which may or may not happen to be correct. I tested it with some colleagues: the derived explicit expression for p(x|beta) was different each time. That also hampered the verification of the subsequent equations, which I eventually abandoned.

b) One very crucial point in the detection of causality for all practical applications is some robustness against false positives, ie. that an algorithm erroneously indicates a causality relation although there is none. How robust is the proposed algorithm in this ubiquitous situation?  What range of odds-ratios is computed and does it match the Jaynes-scale? Here a simple test example with uncorrelated X,Y needs to be added to provide some guidance on the power of the proposed algorithm.

c) The assumption is that the cause distribution follows a lognormal distribution. What are the implications? The lognormal distribution has the range of [0,+infinity[. However, the text states that the range is restricted to [0,1] and that this can always be done. What happens if a single new data point exceeding the previous ones is observed? Are then all points renormalized? This would result in a very unstable algorithm if a single data point result in a shift of all the others and then most likely also the conclusion. Here some clarification is needed.

d) Figure 2 appears to be wrong:

According to the text the right hand graphs display the exponential of the functions given on the left hand side. However, if we compare the blue dotted line and the green dashed line on the left we see that the blue line is always above the green one. However, on the right hand side the corresponding functions intersect each other - something which cannot happen with a monotonous function like exp(x). One could consider perhaps a normalization (as it should be for a probability density) to one - but this is also not the case : integral(p(x)) dx is clearly different from one. Therefore I don't understand what is plotted.

e) FIgure 3: panel a: (x,y)-data pairs are in the range of [0,1]x[0,1]. panel b: histogram of y is given: now in the range of y=0..500 (??). Presumably this is not y but the bin-index. But what is the reason for using this bin-index instead of the correctly binned y? Same holds for panel c.

In conclusion: the paper addresses a topic suited for the journal Entropy - but needs to revised carefully with respect to comprehensibility and embedding into the literature. A assessment of the proposed algorithm with respect to sensitivity (or non-sensitivity) to false positives in the case of no causal relationship between X,Y is crucial.

Author Response

Dear Reviewer,   we want to thank you for your detailed Report. In the following we want to address your issues point by point:

your manuscript 'Bayesian Causal Inference' addresses a relevant topic - how to assess a causal relation from observations only. The paper focusses on the bivariate case. The title of the manuscript is misleading: It raises the impression that the manuscript constitutes a review article on that topic. However, the contribution is a new algorithm to  the problem of causal inference in the special case of two random variables. The title should reflect this.

Following your suggestion, we did in fact decide to adapt the title of our submission, having chosen "A Bayesian Model for Bivariate Causal Inference" now.

Furthermore, precisely the topic of bivariate causal inference has been addressed extensively in the book 'Elements of Causal Inference' (2017) by J. Peters et al. However, this book is mentioned only very briefly in the introduction and not referred to at all in the section on 'problem setting and related work'. This is very unfortunate because the reader would benefit from the more elaborate discussion of the underlying concepts there and could better relate to the manuscript.

We added an additional reference to the title in the section "problem setting and related work" and point the reader to the book for a more detailed discussion.

Which brings me to the main concerns:

a) Some parts of the paper need to be improved significantly on the notation style being used. I got really lost in section 3.1. Which quantities are vectors, which ones are fields, what denotes the range of a parameter, and what are domain definitions, etc. . It may be obvious to the authors but definitely not to the readership. Just an example from section 3.1, line 138: with beta element of R^[0,1] ..some signal field which follows beta~Normal(beta|0,B): First thought: beta is a scalar which takes values between zero and one. Then: beta is a real function on the line [0,1]; next: beta is a Gaussian process. Looking back: wait, there is exp(beta(x))... Being more explicit and precise would spare the reader the decoding process which may or may not happen to be correct. I tested it with some colleagues: the derived explicit expression for p(x|beta) was different each time. That also hampered the verification of the subsequent equations, which I eventually abandoned.

While the set theory-style of denoting a function f:X->Y as an element of Y^X maybe unusual for literature in Bayesian statistics we use it throughout our submission as we consider it an elegant and concise way of defining a functions without adding unnecessary constraints to the functions. We added a footnote (p.5) to clarify the notation.

One very crucial point in the detection of causality for all practical applications is some robustness against false positives, ie. that an algorithm erroneously indicates a causality relation although there is none. How robust is the proposed algorithm in this ubiquitous situation?  What range of odds-ratios is computed and does it match the Jaynes-scale? Here a simple test example with uncorrelated X,Y needs to be added to provide some guidance on the power of the proposed algorithm.

  We consider an additional test on determining a (causal) independence against a causal relation (X->Y or Y->X) to be out of scope for this submission. This could however be a suitable topic for a follow-up research on this topic.

The assumption is that the cause distribution follows a lognormal distribution. What are the implications? The lognormal distribution has the range of [0,+infinity[. However, the text states that the range is restricted to [0,1] and that this can always be done. What happens if a single new data point exceeding the previous ones is observed? Are then all points renormalized? This would result in a very unstable algorithm if a single data point result in a shift of all the others and then most likely also the conclusion. Here some clarification is needed.

In fact we do normalize after having observed all data points. However we first consider all data points and then normalize them. The lognormal distribution is embedded in the Poisson distribution in our model, so the range of [0, +infinity[ is not an issue for our model.

d) Figure 2 appears to be wrong:

According to the text the right hand graphs display the exponential of the functions given on the left hand side. However, if we compare the blue dotted line and the green dashed line on the left we see that the blue line is always above the green one. However, on the right hand side the corresponding functions intersect each other - something which cannot happen with a monotonous function like exp(x). One could consider perhaps a normalization (as it should be for a probability density) to one - but this is also not the case : integral(p(x)) dx is clearly different from one. Therefore I don't understand what is plotted.

Indeed we first apply exp(x) and then perform a normalization (this is the reason for the lines intersecting each other in the one panel and not in the other one). There has however been an issue with the scale in the figure, so the integral was has been different from 1. We fixed this now (and clarified that a normalization is performed)

e) FIgure 3: panel a: (x,y)-data pairs are in the range of [0,1]x[0,1]. panel b: histogram of y is given: now in the range of y=0..500 (??). Presumably this is not y but the bin-index. But what is the reason for using this bin-index instead of the correctly binned y? Same holds for panel c.

We fixed the range for the charts (to [0,1]x[0,1]) now.

In conclusion: the paper addresses a topic suited for the journal Entropy - but needs to revised carefully with respect to comprehensibility and embedding into the literature. A assessment of the proposed algorithm with respect to sensitivity (or non-sensitivity) to false positives in the case of no causal relationship between X,Y is crucial.

  We want to thank you for asserting us the suitability for the Entropy journal. Regarding the sensitivity against False Positives: We in fact think that this is an important issue, however more suitable for a future research paper that extends this submission.

Reviewer 2 Report

The manuscript has presented nice work results that should be published.  Please rearrange paragraphs that have less number of sentences together.  Please check entire paper to make the present better.  

Two-variable causal inference without intervention based on observational data is considered.  One variable,  is called cause and the other,  is called effect.  Assume that both cause and effect are random variables that map onto measurable spaces over a closed finite interval [0, 1] after possible rescaling.   In this work, the cause variable is given by a Poisson lognormal distribution where lognormal distribution addresses the mean of Poisson distribution and by using additive Gaussian noise.  It means that authors proposed a Gaussian Process regression with a Fourier diagonal covariance.  The proposed model allows to delete with the problem of discretization.  Next, Bayesian Causal Inference, BCI, is also proposed for the strategy of Bayesian model selection of the direction of cause effect model.  Algorithm for sampling of causal data via forward model and algorithm for two-variable causal inference using BCI are provided.  Finally, the proposed BCI is compared with current existing models, such as LiNGAM, ANM_HSIC, ANM-MML, IGCI and CGNN through simulated data as well as real world data TCEP.  Generally, the proposed BCI performs reliable with accuracy comparable to the competitors.      

Author Response

Dear Reviewer,

The manuscript has presented nice work results that should be published. Please rearrange paragraphs that have less number of sentences together. Please check entire paper to make the present better.

Thank you for your appreciating review. We concatenated very short paragraphs. We hope you find the updated submission more suitable for reading.

Reviewer 3 Report

This manuscript presents a Bayesian model for testing causal effects, which here are identified with conditional distributions; one first estimates a conditional likelihood of X given Y, then the other way around, and calculates a Bayesian ratio statistic.

The problem is important and there is a significant amount of technical content in the manuscript, but I think the contribution has been overstated. First, on page 2 it is stated that "the possibilities that [the random variables] are...connected by a confounder...are ignored here for the sake of clarity." But this is not an issue of clarity -- rather, it is the foremost practical concern in causal inference. Whenever one deals with observational data, there is always the fear that the estimated relationships between variables are influenced by some sort of unobserved quantity.

Based on my reading, the proposed approach is not able to handle such cases, but rather is focused on Granger-type relationships where one estimates the conditional effects of X on Y and Y on X and makes a final judgment based on which of these seems to be stronger. This is not presented very clearly; for example, on p. 2 the authors refer to "the do-calculus," and it is not made clear how this differs from the usual notion of correlation. The informal description "the intervention do(X=x) can be described as setting the random variable X to attain the value x" makes it sound like P(y|do(x)) is nothing more than the conditional probability P(y|X=x). I believe that most of the model is essentially a hierarchical framework for estimating such probabilities, and the value for actual practical causal inference is not clear to me.

With regard to the numerical experiments, there does appear to be some evidence that the proposed method is competitive (though it does not seem to conclusively outperform the benchmarks on real data), but the model relies on discretization and it was not clear to me how well it would scale to large problems.

Author Response

Dear Reviewer, to respond to your issues:

This manuscript presents a Bayesian model for testing causal effects, which here are identified with conditional distributions; one first estimates a conditional likelihood of X given Y, then the other way around, and calculates a Bayesian ratio statistic.

The problem is important and there is a significant amount of technical content in the manuscript, but I think the contribution has been overstated. First, on page 2 it is stated that "the possibilities that [the random variables] are...connected by a confounder...are ignored here for the sake of clarity." But this is not an issue of clarity -- rather, it is the foremost practical concern in causal inference. Whenever one deals with observational data, there is always the fear that the estimated relationships between variables are influenced by some sort of unobserved quantity.

Based on my reading, the proposed approach is not able to handle such cases, but rather is focused on Granger-type relationships where one estimates the conditional effects of X on Y and Y on X and makes a final judgment based on which of these seems to be stronger.

In fact there is a lot of research considering the same scenario as our paper (deciding only between X->Y and Y->X , see for example the comparative publication "Distinguishing cause from effect using observational data: methods and benchmarks" by Mooij et al., 2014). In our opinion this justifies the focus of the present submission. ( Granger causality is only defined for time series (important here is that the cause observation happens before the effect observation) ).

This is not presented very clearly; for example, on p. 2 the authors refer to "the do-calculus," and it is not made clear how this differs from the usual notion of correlation. The informal description "the intervention do(X=x) can be described as setting the random variable X to attain the value x" makes it sound like P(y|do(x)) is nothing more than the conditional probability P(y|X=x).

In general the probabilities P(y|X=x) and P(y|do(X=x)) are different.  While the conditional probabilty P(y|X=x)  corresponds to just observing the value x  for the variable X, the do-probability P(y|do(X=x)) corresponds to a direct manipulation of the system, only modifying x, without changing any other variable directly.   I believe that most of the model is essentially a hierarchical framework for estimating such probabilities, and the value for actual practical causal inference is not clear to me.  

The model is a hierarchical (bayesian) framework, but not for estimating the above discussed P(y|x) (or P(y|do(x)) for that matter). Instead we want to estimate the probablity of an existing causal mechanism P(X -> Y |d)  conditional to observed data d.

With regard to the numerical experiments, there does appear to be some evidence that the proposed method is competitive (though it does not seem to conclusively outperform the benchmarks on real data), but the model relies on discretization and it was not clear to me how well it would scale to large problems.

In many applications one does have the issue with discrete and sparse data. Especially life sciences which often have to rely on survey data, but also physics, where limited precision of instruments and complex scenarios make it hard to get an abundance of granular observations. But yes, for high-dimensional problems this and many of the competing methods can not be applied. We added a note to the manuscript clarifying this.

Round 2

Reviewer 1 Report

Dear Authors,

the manuscript has been improved. I noticed that it coincides to a large extend with the PhD thesis of the first author. There is nothing wrong with that overlap since a PhD thesis is not peer-reviewed in a strict sense - on the other hand it is a published work on which the present manuscript relies and thus has to be referred to. With this augmentation of the manuscript I have no objections against publication.

Kind regards

Author Response

Dear Reviewer,

we are glad that our revisions are to your agreement. We added a footnote in the abstract that points out the relation to the (master's) thesis.

Best regards

Reviewer 3 Report

The paper has only minor changes from the original submission. The main issues I raised with the previous submission were that 1) the type of causality studied in this paper is very weak, and 2) the proposed analysis is limited to very small problems. Together, these two issues greatly limit the paper's scope and contribution, in my opinion.

Since these issues have not changed in the revised version, my personal view of the paper also remains the same, but I think it is up to the editors to decide whether the new technical content in the paper compensates for the limitations. If this is the case, I think the authors need to make an effort to present a compelling case of an application area where it is valuable to consider this type of causality (without confounding), and where this computationally intensive framework provides practical value. This discussion should not be limited to the one example that is presented, but should try to argue that there is an entire domain where such methods are useful (the authors mention the physical sciences, but I would like to see more specifics).

In addition, I think that the limitations of the paper should be dealt with more transparently, as the title and abstract raise expectations that a very general causal framework will be presented. Regarding the "do-calculus," I appreciate the authors' explanation, but I think it would be better to include a more detailed discussion of these differences in the text itself.

Author Response

Dear reviewer,

regarding your concerns:

The paper has only minor changes from the original submission. The main issues I raised with the previous submission were that 1) the type of causality studied in this paper is very weak, and 2) the proposed analysis is limited to very small problems. Together, these two issues greatly limit the paper's scope and contribution, in my opinion.

Regarding 1), we want to to point out that this restricted type of causal inference is of fundamental nature and a topic of active research (see e.g. "Information-geometric approach to inferring causal directions" by Janzing et. al.) Regarding 2) we would like to suggest once more the possibility to subsample measurements when the amount of data observed becomes a problem (we experienced runtimes of few seconds when considering thousands of observations)

Since these issues have not changed in the revised version, my personal view of the paper also remains the same, but I think it is up to the editors to decide whether the new technical content in the paper compensates for the limitations. If this is the case, I think the authors need to make an effort to present a compelling case of an application area where it is valuable to consider this type of causality (without confounding), and where this computationally intensive framework provides practical value.This discussion should not be limited to the one example that is presented, but should try to argue that there is an entire domain where such methods are useful (the authors mention the physical sciences, but I would like to see more specifics).

We acknowledge that the nature of the exclusively bivariate causal discovery is a rather abstract one and arises mainly in the context of deciding within a subset of possible causal relationships. However there are indeed problems e.g. within astrophysics,  such as fluxes at different wavelength for line of sights through the Milky Way (the Interstellar medium is very complex, with many agents influencing each other, such as magnetic fields, thermal gas, relativistic particles, radiation fields, dust). To have automated ways to detect the main causal relations between those variables would be indeed an advantage. We now do point this out as well in the introduction and the conclusion within our paper.

In addition, I think that the limitations of the paper should be dealt with more transparently, as the title and abstract raise expectations that a very general causal framework will be presented.

We want to point out that we changed the paper's title from "Bayesian Causal Inference" to "A Bayesian Model for Bivariate Causal Inference" in the first revision in order to point out the restriction in the problems considered. We now also added an additional sentence in the abstract to make this restriction more clear.

Regarding the "do-calculus," I appreciate the authors' explanation, but I think it would be better to include a more detailed discussion of these differences in the text itself.

We added the explanation as a footnote in the manuscript

We hope that these considerations as well as the revisions made will find your affirmation und will help to further improve the assessment of our submission.

Best regards,